# Seasonal Effects on Phenolic Contents and In Vitro Health-Promoting Bioactivities of Sacred Lotus (*Nelumbo nucifera*)

**DOI:** 10.3390/plants12071441

**Published:** 2023-03-24

**Authors:** Nattira On-nom, Sirinapa Thangsiri, Woorawee Inthachat, Piya Temviriyanukul, Yuraporn Sahasakul, Chaowanee Chupeerach, Kanchana Pruesapan, Piyapat Trisonthi, Dalad Siriwan, Uthaiwan Suttisansanee

**Affiliations:** 1Food and Nutrition Academic and Research Cluster, Institute of Nutrition, Mahidol University, Salaya, Phuttamonthon, Nakhon Pathom 73170, Thailand; 2Plant Varieties Protection Division, Department of Agriculture, Ministry of Agriculture and Cooperatives, Bangkok 10900, Thailand; 3Institute of Food Research and Product Development, Kasetsart University, Chatuchak, Bangkok 10900, Thailand

**Keywords:** α-glucosidase, angiotensin-converting enzyme, cholinesterase, flower stalk, leaf stalk, lipase, old leaf, petal, seed embryo, stamen

## Abstract

Sacred lotus (*Nelumbo nucifera*) is a commercial product in Asian countries. Almost all parts of the lotus plant are consumed as food or used as traditional medicine due to their high contents of secondary metabolites such as phenolics and alkaloids. However, agricultural management of the sacred lotus occurs during the rainy season, and the plant enters a resting stage during the dry season. Thus, seasonal variation (beginning, middle and end of the rainy season) was investigated for total phenolic contents (TPCs), antioxidant capacities and inhibitions of the key enzymes relevant to chronic diseases including Alzheimer’s disease (β-secretase, acetylcholinesterase and butyrylcholinesterase), hypertension (angiotensin-converting enzyme), obesity (lipase) and diabetes (α-glucosidase) of different sacred lotus parts (seed embryo, petal, stamen, old leaf, leaf stalk and flower stalk). Results indicated that an aqueous extract of stamen in all harvesting seasons exhibited potentially high TPCs, which led to high antioxidant activities and most enzyme inhibitions (up to 53.7-fold higher) than the others collected in the same harvesting period. The phenolic content and biochemical activities in stamen harvested at the beginning of the rainy season were up to 4-fold higher than during other harvesting periods. This information benefits the agricultural management of sacred lotus and supports consumption of different sacred lotus parts for health promotion. Results can be used as an initial database for future product development from different sacred lotus parts.

## 1. Introduction

Sacred lotus (*Nelumbo nucifera*) in the Nelumbonaceae family is an aquatic rhizomatous plant, mainly distributed in Southeast Asia and used as a Buddhism spiritual symbol. In Thailand, sacred lotus parts are used as food ingredients and traditional herbs made from the flower, seed and rhizome. These sacred lotus parts are commonly consumed fresh, while the petal and leaf are steeped in hot water to make a tea beverage. Old leaf, stamen and stalk are rarely consumed due to their hard structure and unattractive appearance. Sacred lotus had been previously reported regarding their nutritional compositions [1,2]. Lotus seed was found to contain 61.3–70.1% carbohydrate, 16.2–28.2% protein and 0.2–3.7% fat [2], while rhizome contained 16.03% carbohydrate, 2.60% protein and 0.10% fat [1]. Temviriyanukul et al. (2020) reported that these plant parts could also be considered as functional ingredients due to their high phenolic contents [3]. Old leaf and stamen exhibited higher total phenolic contents (TPCs) than petal, seed embryo, leaf stalk and flower stalk with flavonoids, especially naringenin, predominantly detected in all sacred lotus parts [3]. Different levels of phenolic acids including *p*-coumaric acid, gallic acid and ferulic acid were also detected. Other than flavonoids and phenolic acids, sacred lotus was also previously reported to contain alkaloids, tannins, saponins, terpenoids and coumarins [4].

Sacred lotus contains a high amount of functional bioactive compounds and is commonly used as a green medicine with pharmaceutical properties such as anti-inflammation, anti-obesity, anti-microbial and anti-Alzheimer’s disease activities depending on the plant parts used [3,4,5]. The fruit and seed have been widely studied regarding their health properties such as the anti-ischemic effect in a rat model [6], hepatoprotective activity in carbon tetrachloride (CCl_4_)- and aflatoxin B1-induced hepatotoxic cell culture [7] and antiproliferative effect in peripheral blood mononuclear cells in human [8]. Leaf was previously reported for its anti-diabetic effect both in vitro and in vivo through inhibition of lipase and α-amylase [9] as well as reducing free cholesterol, total cholesterol and phospholipids in high fat-induced obesity mice [10]. The flowers inhibited hypoglycemic activity in rabbit model [11] and possessed hepatoprotective activity in paracetamol and CCl_4_-induced hepatotoxicity in a rat model [12]. The stamen exhibited high anti-Alzheimer’s disease activities through the inhibition of neurotransmitter degrading enzymes (cholinesterases) and β-amyloid formation enzyme (β-secretase) [3]. The lotus stalk (it not indicated whether this was leaf stalk or flower stalk) also exhibited an anti-pyretic effect in a normal body temperature and yeast-induced pyrexia rat model [13], while the flower stalk was traditionally used to ameliorate bleeding gastric ulcers, post-partum hemorrhage and excessive menstruation [14]. Therefore, unpopular sacred lotus parts with potential health-promoting properties should be further investigated to increase fundamental knowledge on plant consumption and product development.

Sacred lotus is not farmed throughout the year in Thailand because the plant enters a resting stage during the dry season (November–March). Most sacred lotus products are harvested from May to October as early (May–June), middle (July–August) and late (September–October) rainy season. Sacred lotus products collected during different harvesting periods contain varied amounts of bioactive compounds with diverse biochemical activities. Therefore, sample quality control in the sacred lotus industry is difficult, impacting future product development. This study investigated TPCs, antioxidant activities and in vitro health properties of sacred lotus plant parts (seed embryo, petal, stamen, old leaf, leaf stalk and flower stalk) collected at different time periods (beginning, middle and end of the rainy season). In vitro health properties could reduce the risk of non-communicable diseases (NCDs) through inhibition of the key enzymes relevant to Alzheimer’s disease (β-secretase, acetylcholinesterase and butyrylcholinesterase), hypertension (angiotensin-converting enzyme), obesity (lipase) and diabetes (α-glucosidase). Information gained from this study can be used to support agricultural management, consumption and potential product development from different parts of the sacred lotus plant.

## 2. Results

### 2.1. Sample Collection and Harvesting Time

Different sacred lotus parts including the seed embryo, petal, stamen, old leaf, leaf stalk and flower stalk (Figure 1) were received from Kwan Phayao Lotus Community Enterprise, Phayao, Thailand. Samples were collected during November 2017 (time period 1), May 2018 (time period 2) and August 2018 (time period 3) with different rainfall quantity as shown in Table 1. The dry samples were powdered, then subjected to aqueous-based extraction under optimized conditions as previously reported [3] and further investigated for TPCs, antioxidant activities and key enzyme inhibitions.

### 2.2. Total Phenolic Contents

The TPCs of different sacred lotus parts collected at different time periods were investigated using a spectrophotometric assay with Folin–Ciocalteu’s phenol as a reagent and gallic acid as a standard. Results (Table 2) indicated that all sacred lotus parts collected in time period 1 exhibited TPCs ranging 2.75–38.85 gallic acid equivalent (GAE)/g dry weight (DW), while those collected in time period 2 exhibited TPCs ranging 1.31–70.29 mg GAE/g DW. TPCs ranging 0.80–37.82 mg GAE/g DW were found in all samples collected during time period 3. Among all samples, stamen collected during time periods 2 and 3 exhibited higher TPCs than the others. In time period 1, old leaf exhibited the highest TPCs, with stamen second. When comparing different time periods, stamen collected during time period 2 exhibited 1.9-fold higher TPCs than stamen collected at other times, while leaf stalk from all collecting time periods exhibited the lowest TPCs compared to other sacred lotus parts.

### 2.3. Antioxidant Activities

Three antioxidant assays including oxygen radical absorbance capacity (ORAC), ferric ion-reducing antioxidant power (FRAP) and 2,2-diphenyl-1-picrylhydrazyl radical (DPPH•) scavenging assays were performed to investigate the effect of sacred lotus parts and collecting time periods on antioxidant activities. The ORAC assay follows the hydrogen atom transfer (HAT) mechanism of antioxidant ability, while the FRAP and DPPH radical-scavenging assays follow a single-electron transfer (SET) mechanism. Results suggested that all samples collected during time periods 1, 2 and 3 exhibited ORAC activities ranging 119.53–980.53, 130.92–808.44 and 117.88–431.02 µmol TE/g DW, respectively (Table 3). Interestingly, ORAC activity results were corresponded to TPCs. The highest TPCs in old leaf collected during time period 1 yielded the highest ORAC activities, while the highest TPCs in stamen collected during time periods 2 and 3 also provided the highest ORAC activities of stamen in these time periods. Antioxidant activities of stamen in time period 2 were 1.4- and 1.9-fold higher than stamen collected during time periods 1 and 3, respectively. Likewise, the antioxidant activities of old leaf in time period 1 was 1.2- and 2.3-fold higher than old leaf collected in time periods 2 and 3, respectively.

The FRAP activities ranging 15.25–282.92, 29.69–450.77 and 34.19–251.13 µmol TE/g DW were found in all sacred lotus parts collected at time periods 1, 2 and 3, respectively (Table 4). Stamen from all three collecting time periods exhibited the highest FRAP activities, while leaf stalk gave the lowest. Stamen collected in time period 2 exhibited 1.6- and 1.8-fold higher FRAP activities than stamen collected in time periods 1 and 3, respectively.

The DPPH radical-scavenging activities ranged 0.43–1.21, 0.79–1.17 and 0.59–0.93 µmol TE/100 g DW in samples collected at time periods 1, 2 and 3, respectively (Table 4). Unlike FRAP activities with stamen providing the highest, old leaf collected during all time periods exhibited highest DPPH radical-scavenging activities, while stamen was second. Comparing different collecting time periods, old leaf from time periods 1 and 2 exhibited slightly but significantly higher DPPH radical-scavenging activities than old leaf from time period 3.

### 2.4. Enzyme Inhibitory Activities

Alzheimer’s disease (AD) pathogeneses is related to the formation of amyloid plaque generating by degradation of the amyloid precursor protein through the reaction of β-secretase (BACE-1). Deposition of amyloid plaque on the brain causes blockage of oxygen and nutrient exchange between the blood vessels and the brain, leading to AD. Thus, inhibition of BACE-1 is one pathway to reduce the risk of AD. Results indicated that all sacred lotus parts at concentration of 10 mg/mL inhibited BACE-1 activities ranging 51.63–75.34%, 48.42–96.69% and 42.78–97.15% when collecting from time periods 1, 2 and 3, respectively (Table 5). Among sacred lotus parts, stamen collected at time periods 2 and 3 exhibited the highest BACE-1 inhibitions, while stamen was second highest among samples collected at time period 1. Stamen collected at time periods 2 and 3 also exhibited 1.4-fold higher inhibitory activities than stamen at time period 1.

Another hypothesis of AD occurrence is through the cholinergic system consisting of acetylcholinesterase (AChE) and butyrylcholinesterase (BChE), neurotransmitter degradative enzymes. Thus, inhibition of AChE and BChE also reduced the risk of AD. Results indicated that all sacred lotus parts at concentration of 5 mg/mL exhibited AChE inhibitions ranging 42.81–98.15%, 49.66–100.82% and 54.06–99.99% when collecting from time periods 1, 2 and 3, respectively (Table 6). Among all sacred lotus parts, stamen collected at time periods 1 and 3 provided the highest inhibitory activities, while it was in the top three among all samples collected at time period 2. The inhibitory strength of stamen collected at different time periods was slightly but significantly different.

Similar results were also detected in BChE inhibitory assay using extraction concentration of 5 mg/mL. All sacred lotus parts inhibited BChE ranging 24.11–97.51, 36.00–100.61 and 53.49–99.00% when collecting during time periods 1, 2 and 3, respectively (Table 6). Stamen exhibited the highest BChE inhibitory activities among all sacred lotus parts collected in the same time period, while stamen collected at time period 2 exhibited slightly but significantly higher inhibitory activities than stamen collected at the other two time periods.

The renin-angiotensin-aldosterone system (RAAS) has an essential role in maintaining normal blood pressure, while angiotensin-converting enzyme (ACE) in this system functions as angiotensin I degraded enzyme to produce angiotensin II. This product is an octapeptide, functioning as a vasoconstrictor, thereby resulting in increased blood pressure. Thus, ACE inhibition controlled and reduced the risk of hypertension. Results indicated that all sacred lotus parts at concentration of 0.5 mg/mL collected from time periods 1, 2 and 3 exhibited ACE inhibitions ranging 21.38–94.91, 29.78–98.62 and 44.57–89.04%, respectively (Table 7). Among all sacred lotus parts, stamen collected during time periods 2 and 3 exhibited the highest ACE inhibitory activities. Old leaf provided the highest inhibition among all samples collected in time period 1, while stamen was in second place. When comparing different collecting time periods, stamen in time period 2 exhibited 1.1-fold higher ACE inhibitory activities that the others, while old leaf collected in time period 1 exhibited 1.8- to 2.0-fold higher ACE inhibition than the rest.

Type 2 diabetes mellitus (T2DM) is caused by the combination of insulin resistance and insulin secretion deficiency from constant high plasma glucose. Thus, retardment of carbohydrate degradation into glucose by inhibiting α-glucosidase reaction is one of the medicinal targets of the T2DM synthetic drug, acarbose. Results indicated that all sacred lotus parts at concentration of 10 mg/mL inhibited α-glucosidase activities ranging 24.17–86.86%, 32.01–91.84% and 29.07–93.89% when collecting from time periods 1, 2 and 3, respectively (Table 8). Among the sacred lotus parts, stamen collected from all time periods exhibited the highest α-glucosidase inhibition, while stamen collected at time periods 2 and 3 exhibited 1.1-fold higher inhibitory activities than stamen at time period 1.

Overweight and obesity is caused by an energy imbalance between calorie intake and calorie expenditure. Consumption of high-fat foods is one of the major risk factors for overweight and obesity; thus, reduced fat absorption by inhibition of the lipid degrading enzyme, lipase, is one of the medicinal targets for the synthetic drug, orlistat, as well as other natural products. Results indicated that all sacred lotus parts at concentration of 5 mg/mL exhibited lipase inhibition ranging 3.59–86.78, 11.11–98.01 and 6.25–71.14% when collecting from time periods 1, 2 and 3, respectively (Table 9). Unlike other properties with stamen providing high inhibitory strength, old leaf from time period 1 and petal from time periods 2 and 3 exhibited the highest lipase inhibitory activities among all samples collected in the same time period. Old leaf collected at time period 1 exhibited 3.3- to 3.9-fold higher lipase inhibition that the others, while lipase inhibition of petal from time period 2 was 3.7- and 1.4-fold higher than petal from time periods 1 and 3, respectively.

### 2.5. Principal Component Analysis

This study demonstrated numerous and complicated data for various sacred lotus parts acquired from different growing seasons. Therefore, Principal Component Analysis (PCA) was performed as a statistical tool to unravel the data and reach conclusions. Mean values of TPCs, antioxidant activities determined by ORAC, FRAP and DPPH radical-scavenging assays, and inhibition of the key enzymes relevant to chronic diseases including BACE-1, AChE, BChE, ACE, α-glucosidase and lipase (Table 2, Table 3, Table 4, Table 5, Table 6, Table 7, Table 8 and Table 9) were used in the PCA analysis. Figure 2 shows biplot data containing active variables in red (TPCs, antioxidant capacities (ORAC, FRAP and DPPH radical-scavenging activities), enzyme inhibitory activities against BACE-1, AChE, BChE, ACE, α-glucosidase and lipase) and active observations in blue (seed embryo (SE), petal (PE), old leaf (OL), stamen (ST), flower stalk (FS) and leaf stalk (LS) harvested at different time periods). Two PCs, PC1 and PC2, covering 56.12% and 14.89% of all variables, respectively, for a total of 71.01%, indicate good data representability. The biplot clearly shows a scattered pattern, indicating the absence of relationships between lotus parts. Conversely, all active variables, except BACE-1, were clustered together implying some correlation. BACE-1 had a negative correlation with α-glucosidase, implying that more BACE-1 inhibitory activity correlated with low α-glucosidase inhibitory activity. Further detailed investigations on phytochemical contents or antagonist effects are required.

The biplot also shows that stamen harvested from all three time periods (ST1, ST2 and ST3), old leaf harvested from time period 1 (OL1) and petal harvested from time periods 2 and 3 (PE2 and PE3, respectively) exhibited high TPCs as well as antioxidant and enzyme inhibitory activities. They were clustered together (Figure 2), confirming the data in Table 2, Table 3, Table 4, Table 5, Table 6, Table 7, Table 8 and Table 9, while seed embryo, leaf stalk and flower stalk harvested in any season revealed low TPCs, antioxidant activities and enzyme inhibitions. These data indicated that minor effects of growing seasons were noted for stamen, even though time period 2 was the most suitable climate to collect this sacred lotus part. Unclear effects of growing seasons were observed in leaf stalk, flower stalk, old leaf and petal as they were scattered throughout the biplot. In conclusion, utilization of stamen for health benefits is practical in all seasons because stamen yielded higher TPCs and most health-promoting bioactivities than other sacred lotus parts collected during the same time period. Other sacred lotus parts require specific harvesting time periods, especially old leaf and petal. The PCA analysis suggested that old leaf should be collected during time period 1 (OL1), while petal should be collected during time periods 2 and 3 (PE2 and PE3, respectively) for maximal health-promoting bioactivities.

## 3. Discussion

Several sacred lotus parts have well-recognized pharmacological activities, resulting from their contained phytochemicals. Some sacred lotus parts, i.e., seed, leaf and root have been widely studied and used as both general foods and green medicine, while other parts, i.e., petal, stamen and stalk are underutilized due to limited information on their bioactive compounds and health-promoting activities [5]. Sacred lotus harvested at different time periods has variegated biological properties due to diversity of the bioactive compounds. In this study, different parts of sacred lotus including seed embryo, petal, stamen, old leaf, leaf stalk and flower stalk collected at different time periods were subjected to aqueous-based extraction and investigated regarding their TPCs, antioxidant activities and key enzyme inhibitions relevant to AD (BACE-1, AChE and BChE), hypertension (ACE), diabetes (α-glucosidase) and obesity (lipase). The results indicated that (i) stamen exhibited potentially high TPCs, leading to high antioxidant activities and most enzyme inhibitions (BACE-1, AChE, BChE, ACE and α-glucosidase), (ii) petal and old leaf exhibited high lipase inhibition despite having low TPCs and (iii) these activities were superior in stamen collected during time period 2 (or at the beginning of the rainy season) than the other time periods.

The previous study had identified phenolics in different sacred lotus parts by using high-performance liquid chromatography (HPLC) [3]. It was found that naringenin, cyanidin and delphinidin were predominantly detected in seed embryo and petal, while flower stalk, leaf stalk and stamen exhibited exceptionally high naringenin contents [3]. Besides, old leaf also contained high naringenin, quercetin and cyanidin [3]. Among sacred lotus parts, we found that stamen exhibited relatively high TPCs, corresponding to the previous findings of high TPCs in both stamen and old leaf [3], leading to high antioxidant activities. In this study, stamen collected during every harvesting period exhibited highest FRAP activities, while ORAC activities varied according to sacred lotus part and harvesting time periods. A previous study indicated that even though stamen and old leaf exhibited similar TPCs (7% variation), stamen exhibited 1.2-fold higher FRAP activities, while old leaf exhibited 1.8-fold higher ORAC activities [3]. This information suggested that antioxidative agents in stamen preferred the SET-based mechanism (FRAP assay), while those of old leaf preferred the HAT-based mechanism (ORAC assay). Stamen contained *p*-coumaric acid, myricetin, quercetin, naringenin, kaempferol, isorhamnetin, cyanidin and delphinidin, with naringenin being the most abundant phenolic detected by (HPLC [3]. The density functional theory determined that naringenin in water phase preferred a sequential proton loss electron transfer (SPLET) owing to its 7−OH moiety (the hydroxyl group attached to the position of C7 in naringenin structure) to HAT- and SET-based mechanisms [15]. This SPLET mechanism included two steps of proton transfer from the antioxidant after electron donation from the anion. 

High TPCs in stamen also led to high BACE-1, AChE, BChE, ACE and α-glucosidase inhibitory activities. These results concurred with a previous study, indicating the lowest half maximal inhibitory concentration (IC_50_) of stamen against AChE and BChE among all six sacred lotus parts, while BACE-1 inhibition showed 6.5% variation, with highest activities detected in flower stalk and leaf stalk [3]. Naringenin inhibited BACE-1 with IC_50_ of 30.31 µM, AChE with IC_50_ of 42.66 µM and BChE with IC_50_ of >100 µM [16], showing lower effectiveness than the synthetic drug, donepezil (IC_50_ values of 1.31, 3.12 and 2.14 µM against BACE-1, AChE and BChE, respectively) [17]. Naringenin (500 µM) also inhibited ACE at 22.3% [18]. Interestingly, a previous in vivo experiment indicated that naringenin (50 mg/kg) downregulated ACE and mineralocorticoid receptor (indicators of hypertension) and also alleviated neuronal oxidative stress (indicator of neurological damage) in L-N^G^-nitro arginine methyl ester (L-NAME)-induced hypertensive rat [19]. These results indicated that naringenin possessed neuroprotective and hypertensive inhibitory effects even though the in vitro enzyme inhibitory investigation suggested that naringenin was not a good inhibitor. Naringenin was also an effective inhibitor against α-glucosidase (IC_50_ of 0.79 mM) comparing to acarbose, an anti-diabetic drug (IC_50_ of 2.03 mM) [20]. Naringenin (25 mg/kg) could lower postprandial serum glucose levels in streptozotocin (STZ)-induced diabetic rats [21]. These data suggested that stamen with potentially high naringenin content showed promise as an excellent provider for AD, hypertensive and diabetic inhibitory agents.

Despite having low TPCs, petal and old leaf exhibited high lipase inhibition. Sacred lotus leaf has a long history of use as an anti-obesity agent in both traditional and modern medicines [4]. Two quercetin derivatives, quercetin-3-*O*-*β*-D-arabinopyranosyl-(1→2)-*β*-D-galactopyranoside and quercetin-3-*O*-*β*-D-glucuronide in sacred lotus leaf inhibited lipase with the IC_50_ of 88.8 and 35.8 µM, respectively [22]. However, compared to orlistat, a commercially available lipase inhibitor with an IC_50_ value of 1.53 µM [22], these two flavonoids were less effective. Nevertheless, leaf extract (140–560 mg/kg) could reduce total cholesterol (TC) and triglycerides (TG) in egg yolk-induced acute hyperlipidemia rat [8], while it also lowered TC, TG and low-density lipoprotein cholesterol (LDL-C) levels, and increased high-density lipoprotein cholesterol (HDL-C) level in high fat diet-induced hyperlipidemia rat [8]. Previous research mainly focused on sacred lotus leaf, with scant information on petal. It was previously reported that petal exhibited inhibitory effect on lipid storage in adipocytes and promoted lipolysis by inhibiting lipase with the IC_50_ of 47 µg/mL [23].

Variation in harvesting time also affected TPCs and biochemical activities. Stamen, collected at the beginning of the rainy season exhibited higher TPCs, antioxidant activities and enzyme inhibitions than stamen collected from the other two time periods. Our results concurred with a report on *Adenia viridiflora* Craib., indicating that higher TPCs were obtained in plants collected during March and April, after the plant resting stage [24]. Low rainfall is experienced in March in Thailand, with increased precipitation during the late April monsoon season. Thus, high phenolic biosynthesis occurs after the plant resting stage or at the beginning of the rainy season. Phenolics with numerous biological functions (i.e., anti-microorganism and antioxidant activities) may also benefit plant growth and survival [24,25].

## 4. Materials and Methods

### 4.1. Sample Preparation and Extraction

Dry sacred lotus parts including seed embryo, petal, stamen, old leaf, leaf stalk and flower stalk were obtained from Kwan Phayao Lotus Community Enterprise, Phayao, Thailand in November 2017 (time period 1), May 2018 (time period 2) and August 2018 (time period 3). The mean annual temperature (highest/lowest) and mean annual precipitation in 2017 were 31.7/20.9 °C and 3.7 mm, respectively, while mean annual temperature (highest/lowest) and mean annual precipitation in 2018 were 31.5/20.9 °C and 3.5 mm, respectively (the data from the Northern Meteorological Center, Thai Meteorological Department, Chiang Mai, Thailand (http://www.cmmet.tmd.go.th/station/phayao/, accessed on 2 February 2023). The samples were submitted to Sireeruckhachati Nature Learning Park, Mahidol University (Nakhon Pathom, Thailand) for identification and registered for voucher specimen (PBM-005675 for old leaf and leaf stalk and PBM-006071 for petal, stamen, seed embryo and flower stalk). All samples were ground using a grinder (Philips 600 W series from Philips Electronics Co., Ltd., Jakarta, Indonesia) into a fine powder before packing in vacuum aluminum foil bags and kept at −20 °C until further analysis. 

Physical characterizations of powdered samples were investigated regarding moisture content and color using a Halogen Moisture Analyzer (HE53 series from Mettler-Toledo AG, Greifensee, Switzerland) and a ColorFlex EZ Spectrophotometer (Hunter Associates Laboratory, Reston, VA, USA), respectively, with results shown in Appendix A, respectively.

An aqueous-based extraction of all samples was performed as previously reported [3]. Briefly, a powdered sample (0.1 g) was mixed in ultrapure water (10 mL) and incubated in a water bath shaker (WNE45 series from Memmert GmBh, Eagle, WI, USA) at 90 °C for one hour. The supernatant was collected after centrifugation for 15 min at 3800× *g* using a Hettich^®^ Rotina 38R refrigerated centrifuge (Andreas Hettich GmbH, Tuttlingen, Germany). All sacred lotus extracts were filtered through a 0.45 µM PES membrane syringe filter and kept at −20 °C until required for further analyses.

### 4.2. Determination of Total Phenolic Contents (TPCs)

The TPCs of all sacred lotus extracts from Section 4.1 were determined using Folin–Ciocalteu’s phenol reagent and gallic acid (0–200 µg/mL) as a standard. The absorbance was read at 765 nm. The procedure followed a well-established protocol as previously reported without any modification [26]. The reactions were performed on a Synergy^TM^ HT 96-well UV-Vis microplate reader and Gen 5 data analysis software (BioTek Instruments, Inc., Winooski, VT, USA). All chemicals and reagents were purchased from Sigma-Aldrich (St. Louis, MO, USA).

### 4.3. Determination of Antioxidant Activities

Antioxidant activities of all sacred lotus extracts from Section 4.1 were determined using three antioxidant assays including ORAC, FRAP and DPPH radical-scavenging assays as previously described without any modification [26]. Briefly, the ORAC assay used 2,2′-azobis(2-amidinopropane) dihydrochloride and sodium fluorescein as the main reagents with kinetical detection at 485 nm excitation wavelength and 528 nm emission wavelength. The FRAP assay consisted of FRAP reagent (2,4,6-tri(2-pyridyl)-s-triazine, acetate buffer and FeCl_3_·6H_2_O solution) with endpoint detection at 600 nm, while the DPPH radical-scavenging assay employed DPPH radical reagent and endpoint detection at 520 nm. Trolox was used as a standard, and all reactions were performed using a microplate reader. All chemicals and reagents were purchased from Sigma-Aldrich (St. Louis, MO, USA).

### 4.4. Enzyme Inhibitory Assay 

Inhibitory assays on the key enzymes relevant to AD (BACE-1, AChE and BChE), hypertension (ACE), obesity (lipase) and diabetes (α-glucosidase) were performed according to the well-established protocols as previously reported [27,28,29]. Enzyme assay components including enzymes, chemicals and reagents are summarized in Table 10. All assay reactions were visualized using the microplate reader.

The results were expressed as the percentage of inhibition (% inhibition) using the Equation (1) as follows,
(1)$$\%\;\mathrm{inhibition}=\left(1-\frac{B-b}{A-a}\right)\;\times \;100,$$
where *A* is an initial velocity (*V*_0_) of the control reaction with an enzyme (a control), *a* is an *V*_0_ of the control reaction without enzyme (a control blank), *B* is an *V*_0_ of the enzyme reaction with extract (a sample), and *b* is an *V*_0_ of the reaction with extract but without enzyme (a sample blank). All the enzymes, chemicals, and reagents in the enzyme inhibitory assays were purchased from Sigma-Aldrich (St. Louis, MO, USA).

### 4.5. Statistical Analysis

All experiments were carried out in triplicate (*n* = 3), with results expressed as the mean ± standard deviation (SD). Significant difference at *p* < 0.05 was determined by one-way analysis of variance (ANOVA) and Duncan’s multiple comparison test. PCA of TPCs, antioxidant capacities and enzyme inhibitory activities were analyzed utilizing XLSTAT^®^ (Addinsoft Inc., New York, NY, USA).

## 5. Conclusions

Limited information on phytochemical contents and health-promoting activities of some underutilized sacred lotus parts may lead to new product development following the zero food waste concept. Suitable plant-harvesting periods could improve agricultural management and achieve optimized benefits both for direct consumption and future product development (i.e., supplements and active ingredients). This study investigated TPCs, antioxidant activities and inhibition of key enzymes relevant to NCD occurrence. Results indicated that stamen provided high TPCs as well as antioxidant and enzyme inhibitory activities among all investigated sacred lotus parts, with optimal results in stamen harvested at the beginning of the rainy season or after the plant resting stage. This information can be used to promote consumption and product development of different parts of the sacred lotus plant collected at the most suitable harvesting time. However, further cell culture or in vivo experiments are required to confirm our in vitro results.

## Figures and Tables

**Figure 1 plants-12-01441-f001:**
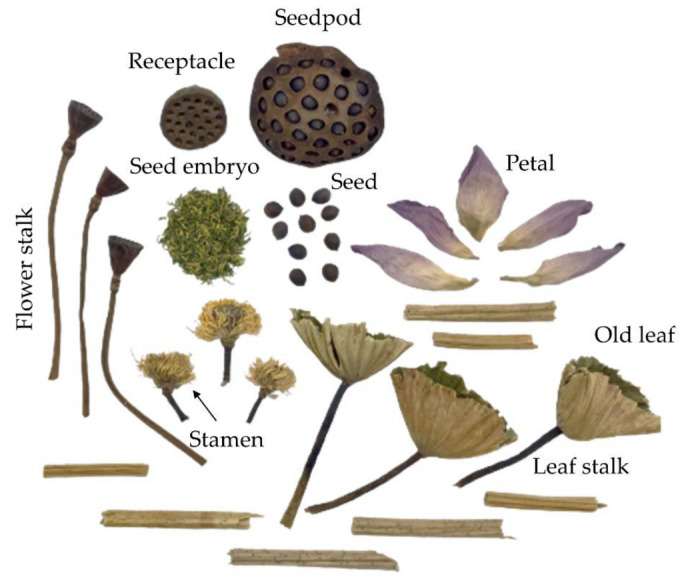
Different sacred lotus parts used in this experiment including seed embryo, petal, stamen, old leaf, leaf stalk and flower stalk.

**Figure 2 plants-12-01441-f002:**
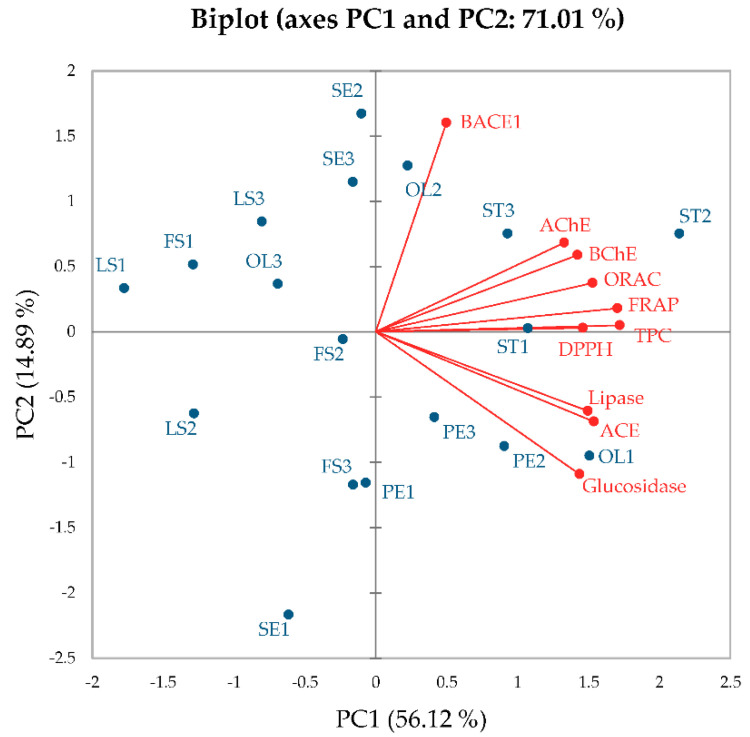
Biplot of principal component analysis (PCA) generated from the observations including seed embryo (SE), petal (PE), old leaf (OL), stamen (ST), flower stalk (FS) and leaf stalk (LS) harvested from different time periods (1: time period 1; 2: time period 2; 3: time period 3) and variables including total phenolic contents (TPCs), antioxidant activities (FRAP, DPPH radical-scavenging and ORAC activities) and enzyme inhibitory activities against β-secretase (BACE-1), acetylcholinesterase (AChE), butyrylcholinesterase (BChE), angiotensin-converting enzyme (ACE), α-glucosidase and lipase.

**Table 1 plants-12-01441-t001:** Statistics of rainfall at Ban Tom, Mueang Phayao District, Phayao province, Thailand.

Time Period	Harvesting Times (Month/Year)	Season in Thailand	Total/Mean/Range of Rainfall (mm)
1	November/2017	End of rainy season	50.0/1.7/0.0–24.1
2	May/2018	Beginning of rainy season	157.0/5.1/0.0–34.6
3	August/2018	Middle of rainy season	165.2/5.3/0.0–72.1

Statistics of rainfall was received from Northern Meteorological Center, Thai Meteorological Department, Chiang Mai, Thailand (http://www.cmmet.tmd.go.th/station/phayao/, accessed on 2 February 2023).

**Table 2 plants-12-01441-t002:** Total phenolic contents (TPCs) of different sacred lotus parts collected from different time periods.

Plant Parts	Total Phenolic Contents (mg GAE/g DW)		
	Time 1	Time 2	Time 3
Seed embryo	12.78 ± 0.15 ^cB^	14.62 ± 1.40 ^cdA^	12.45 ± 1.23 ^bB^
Stamen	36.28 ± 0.69 ^bB^	70.29 ± 6.26 ^aA^	37.82 ± 2.94 ^aB^
Old leaf	38.85 ± 0.51 ^aA^	5.11 ± 0.62 ^deC^	6.84 ± 0.60 ^dB^
Petal	12.11 ± 0.15 ^cB^	26.62 ± 2.19 ^bA^	13.53 ± 0.82 ^bB^
Flower stalk	4.30 ± 0.08 ^dC^	19.00 ± 0.73 ^bcA^	10.53 ± 0.74 ^cB^
Leaf stalk	2.75 ± 0.07 ^eA^	1.31 ± 0.14 ^eB^	0.80 ± 0.05 ^eC^

All data are demonstrated as the mean ± standard deviation (SD) of triplicate experiments (*n* = 3). Lowercase letters indicate significantly different TPCs of different sacred lotus parts collecting in the same time period, while uppercase letters indicate significantly different TPCs of the same sacred lotus part collecting at different time periods at *p* < 0.05 using one-way analysis of variance (ANOVA) and Duncan’s multiple comparison test. DW: dry weight; GAE: gallic acid equivalent.

**Table 3 plants-12-01441-t003:** Antioxidant activities measured by oxygen radical absorbance capacity (ORAC) assay of different sacred lotus parts collected from different time periods.

Plant Parts	ORAC Activities (µmol TE/g DW)		
	Time 1	Time 2	Time 3
Seed embryo	294.87 ± 5.30 ^dC^	682.60 ± 44.83 ^bA^	585.82 ± 26.50 ^aB^
Stamen	568.02 ± 7.17 ^bB^	808.44 ± 62.75 ^aA^	431.02 ± 29.51 ^bC^
Old leaf	980.53 ± 29.28 ^aA^	687.76 ± 57.31 ^bB^	322.44 ± 31.03 ^cC^
Petal	375.10 ± 12.48 ^cB^	577.14 ± 33.74 ^cA^	338.53 ± 28.69 ^cB^
Flower stalk	207.22 ± 6.83 ^eA^	130.92 ± 11.30 ^eB^	220.41 ± 5.06 ^dA^
Leaf stalk	119.53 ± 3.10 ^fB^	187.27 ± 16.13 ^dA^	117.88 ± 10.80 ^eB^

All data are demonstrated as the mean ± standard deviation (SD) of triplicate experiments (*n* = 3). Lowercase letters indicate significantly different antioxidant activities of different sacred lotus parts collected in the same time period, while uppercase letters indicated significantly different antioxidant activities of the same sacred lotus part collected at different time periods at *p* < 0.05 using one-way analysis of variance (ANOVA) and Duncan’s multiple comparison test. DW: dry weight; TE: Trolox equivalent.

**Table 4 plants-12-01441-t004:** Antioxidant activities measured by ferric ion-reducing antioxidant power (FRAP) and 2,2-diphenyl-1-picrylhydrazyl (DPPH) radical-scavenging assays of different sacred lotus parts collected from different time periods.

Plant Parts	FRAP Activities (µmol TE/g DW)			DPPH Radical-Scavenging Activities (µmol TE/100 g DW)		
	Time 1	Time 2	Time 3	Time 1	Time 2	Time 3
Seed embryo	81.95 ± 1.49 ^cB^	58.24 ± 3.31 ^dC^	88.20 ± 3.09 ^bcA^	0.73 ± 0.01 ^cA^	0.79 ± 0.07 ^dA^	0.76 ± 0.06 ^cA^
Stamen	282.92 ± 2.03 ^aB^	450.77 ± 16.93 ^aA^	251.13 ± 20.27 ^aC^	1.07 ± 0.01 ^bA^	1.12 ± 0.04 ^bA^	0.81 ± 0.05 ^bcB^
Old leaf	230.69 ± 1.41 ^bA^	98.27 ± 3.29 ^cB^	83.02 ± 7.95 ^cC^	1.21 ± 0.00 ^aA^	1.17 ± 0.02 ^aA^	0.93 ± 0.08 ^aB^
Petal	70.79 ± 1.69 ^dC^	154.92 ± 5.00 ^bA^	97.02 ± 5.54 ^bB^	0.72 ± 0.03 ^cB^	0.94 ± 0.08 ^cA^	0.86 ± 0.06 ^bA^
Flower stalk	23.13 ± 0.13 ^eC^	29.69 ± 2.89 ^eB^	38.78 ± 3.53 ^dA^	0.62 ± 0.03 ^dC^	0.91 ± 0.03 ^cA^	0.76 ± 0.04 ^cB^
Leaf stalk	15.25 ± 0.18 ^fC^	30.12 ± 2.44 ^eB^	34.19 ± 2.29 ^dA^	0.43 ± 0.01 ^eC^	0.89 ± 0.06 ^cA^	0.59 ± 0.04 ^dB^

All data are demonstrated as the mean ± standard deviation (SD) of triplicate experiments (*n* = 3). Lowercase letters indicate significantly different antioxidant activities of different sacred lotus parts collected in the same time period, while uppercase letters indicated significantly different antioxidant activities of the same sacred lotus part collected at different time periods at *p* < 0.05 using one-way analysis of variance (ANOVA) and Duncan’s multiple comparison test. DW: dry weight; TE: Trolox equivalent.

**Table 5 plants-12-01441-t005:** **β**-Secretase (BACE-1) inhibitory activities of different sacred lotus parts collected from different time periods.

Plant Parts	BACE-1 Inhibitory Activities (% Inhibition) *		
	Time 1	Time 2	Time 3
Seed embryo	45.53 ± 2.31 ^dC^	82.14 ± 2.10 ^bA^	75.09 ± 2.01 ^bcB^
Stamen	71.14 ± 0.35 ^bB^	96.69 ± 0.61 ^aA^	97.15 ± 3.47 ^aA^
Old leaf	55.62 ± 1.47 ^cB^	78.44 ± 1.39 ^bA^	79.28 ± 1.43 ^bA^
Petal	51.63 ± 0.93 ^cB^	56.75 ± 2.75 ^cB^	62.93 ± 3.17 ^dA^
Flower stalk	75.20 ± 0.70 ^aA^	51.99 ± 8.42 ^cdB^	42.78 ± 2.93 ^eB^
Leaf stalk	75.34 ± 2.48 ^aA^	48.42 ± 1.12 ^dB^	71.86 ± 2.57 ^cA^

All data are demonstrated as the mean ± standard deviation (SD) of triplicate experiments (*n* = 3). Lowercase letters indicate significantly different inhibitory activities of different sacred lotus parts collected in the same time period, while uppercase letters indicated significantly different inhibitory activities of the same sacred lotus part collected at different time periods at *p* < 0.05 using one-way analysis of variance (ANOVA) and Duncan’s multiple comparison test. * Final extract concentration = 10 mg/mL.

**Table 6 plants-12-01441-t006:** Acetylcholinesterase (AChE) and butyrylcholinesterase (BChE) inhibitory activities of different sacred lotus parts collected from different time periods.

Plant Parts	AChE Inhibitory Activities (% Inhibition) *			BChE Inhibitory Activities (% Inhibition) *		
	Time 1	Time 2	Time 3	Time 1	Time 2	Time 3
Seed embryo	47.45 ± 0.71 ^eC^	100.82 ± 0.68 ^aA^	97.30 ± 1.64 ^bB^	24.11 ± 0.46 ^fC^	98.88 ± 0.64 ^aA^	87.75 ± 3.95 ^bB^
Stamen	98.15 ± 1.45 ^aB^	98.70 ± 0.90 ^bcB^	99.99 ± 0.35 ^aA^	97.51 ± 0.59 ^aC^	100.61 ± 0.73 ^aA^	99.00 ± 1.28 ^aB^
Old leaf	87.58 ± 1.55 ^bB^	97.68 ± 3.93 ^cA^	54.06 ± 3.05 ^eC^	89.17 ± 2.21 ^bB^	99.98 ± 4.81 ^aA^	53.49 ± 3.50 ^cC^
Petal	81.14 ± 0.30 ^cC^	97.28 ± 0.51 ^cB^	98.97 ± 0.52 ^abA^	77.78 ± 0.38 ^cC^	98.41 ± 0.28 ^aA^	95.26 ± 2.59 ^abB^
Flower stalk	68.03 ± 3.49 ^dC^	100.09 ± 0.28 ^abA^	91.39 ± 3.48 ^dB^	31.93 ± 1.75 ^dC^	99.38 ± 1.82 ^aA^	93.21 ± 1.37 ^abB^
Leaf stalk	42.81 ± 1.53 ^fC^	49.66 ± 2.77 ^dB^	94.33 ± 1.58 ^cA^	26.94 ± 0.55 ^eC^	36.00 ± 3.26 ^bB^	90.37 ± 2.66 ^abA^

All data are demonstrated as the mean ± standard deviation (SD) of triplicate experiments (*n* = 3). Lowercase letters indicate significantly different inhibitory activities of different sacred lotus parts collected in the same time period, while uppercase letters indicated significantly different inhibitory activities of the same sacred lotus part collected at different time periods at *p* < 0.05 using one-way analysis of variance (ANOVA) and Duncan’s multiple comparison test. * Final extract concentration = 5 mg/mL.

**Table 7 plants-12-01441-t007:** Angiotensin-converting enzyme (ACE) inhibitory activities of different sacred lotus parts collected from different time periods.

Plant Parts	ACE Inhibitory Activities (% Inhibition) *		
	Time 1	Time 2	Time 3
Seed embryo	87.31 ± 0.24 ^cA^	50.91 ± 1.32 ^cB^	44.57 ± 3.04 ^eC^
Stamen	91.98 ± 0.11 ^bB^	98.62 ± 0.84 ^aA^	89.04 ± 1.66 ^aC^
Old leaf	94.91 ± 0.02 ^aA^	48.03 ± 4.48 ^cB^	51.93 ± 3.35 ^dB^
Petal	91.34 ± 0.05 ^bA^	69.65 ± 6.71 ^bB^	72.56 ± 7.07 ^bB^
Flower stalk	25.38 ± 0.44 ^dC^	37.89 ± 3.77 ^dB^	63.05 ± 4.12 ^cA^
Leaf stalk	21.38 ± 0.44 ^eC^	29.78 ± 0.87 ^eB^	60.08 ± 4.01 ^cA^

All data are demonstrated as the mean ± standard deviation (SD) of triplicate experiments (*n* = 3). Lowercase letters indicate significantly different inhibitory activities of different sacred lotus parts collected in the same time period, while uppercase letters indicated significantly different inhibitory activities of the same sacred lotus part collected at different time periods at *p* < 0.05 using one-way analysis of variance (ANOVA) and Duncan’s multiple comparison test. * Final extract concentration = 0.5 mg/mL.

**Table 8 plants-12-01441-t008:** α-Glucosidase inhibitory activities of different sacred lotus parts collected from different time periods.

Plant Parts	α-Glucosidase Inhibitory Activities (% Inhibition) *		
	Time 1	Time 2	Time 3
Seed embryo	86.86 ± 0.10 ^aA^	32.01 ± 2.76 ^eB^	31.65 ± 2.86 ^eB^
Stamen	86.31 ± 1.12 ^aB^	91.84 ± 2.32 ^aA^	93.89 ± 1.21 ^aA^
Old leaf	85.78 ± 0.27 ^aA^	45.41 ± 2.02 ^bB^	43.40 ± 2.41 ^dB^
Petal	84.09 ± 0.27 ^bB^	89.35 ± 3.45 ^bA^	88.56 ± 0.83 ^bA^
Flower stalk	37.08 ± 0.69 ^cC^	51.09 ± 2.22 ^cB^	83.67 ± 7.33 ^cA^
Leaf stalk	24.17 ± 0.54 ^dC^	32.76 ± 1.91 ^eA^	29.07 ± 2.61 ^eB^

All data are demonstrated as the mean ± standard deviation (SD) of triplicate experiments (*n* = 3). Lowercase letters indicate significantly different inhibitory activities of different sacred lotus parts collected in the same time period, while uppercase letters indicated significantly different inhibitory activities of the same sacred lotus part collected at different time periods at *p* < 0.05 using one-way analysis of variance (ANOVA) and Duncan’s multiple comparison test. * Final extract concentration = 10 mg/mL.

**Table 9 plants-12-01441-t009:** Lipase inhibitory activities of different sacred lotus parts collected from different time periods.

Plant Parts	Lipase Inhibitory Activities (% Inhibition) *		
	Time 1	Time 2	Time 3
Seed embryo	18.37 ± 1.44 ^dB^	11.11 ± 0.66 ^eC^	36.31 ± 3.13 ^cA^
Stamen	29.27 ± 0.68 ^bB^	70.63 ± 4.82 ^bA^	34.44 ± 3.36 ^cB^
Old leaf	86.78 ± 2.82 ^aA^	26.24 ± 1.74 ^dB^	9.77 ± 0.59 ^dC^
Petal	26.28 ± 0.86 ^cC^	98.01 ± 3.43 ^aA^	71.14 ± 2.87 ^aB^
Flower stalk	5.71 ± 0.36 ^eB^	44.87 ± 3.26 ^cA^	42.27 ± 2.68 ^bA^
Leaf stalk	3.59 ± 0.24 ^eC^	11.58 ± 1.06 ^eA^	6.25 ± 0.48 ^eB^

All data are demonstrated as the mean ± standard deviation (SD) of triplicate experiments (*n* = 3). Lowercase letters indicate significantly different inhibitory activities of different sacred lotus parts collected in the same time period, while uppercase letters indicated significantly different inhibitory activities of the same sacred lotus part collected at different time periods at *p* < 0.05 using one-way analysis of variance (ANOVA) and Duncan’s multiple comparison test. * Final extract concentration = 5 mg/mL.

**Table 10 plants-12-01441-t010:** The assay components including an enzyme, a substrate, an indicator, a fruit extract and a detection wavelength for enzyme inhibitory assays.

Assay	Assay Components				
	Enzyme	Substrate	Indicator	Extract	Detection Wavelength
BACE-1	BACE-1 FRET assay kit (Sigma-Aldrich, St. Louis, MO, USA) using manufacturer’s recommendations				λ_ex_ = 320 nm λ_em_ = 405 nm
AChE	100 μL of 0.05 µg/mL AChE ^1^	50 μL of 0.32 mM ACh	10 µL of 16 mM DTNB	40 µL	412 nm
BChE	100 μL of 0.5 µg/mL BChE ^2^	50 μL of 0.4 mM BCh			
Lipase	100 µL of 10 µg/mL lipase ^3^	50 μL of 0.2 mM DMPTB			
ACE	3 µL of 0.5 U/mL ACE ^4^	30 µL of 3 mM HHL	15 µL of 20 mg/mL OPA	50 µL	λ_ex_ = 360 nm λ_em_ = 485 nm
α-Glucosidase	100 µL of 0.1 U/mL α-glucosidase ^5^	50 µL of 2 mM pNPG		50 µL	405 nm

BACE-1: β-secretase; FRET: fluorescence resonance energy transfer; AChE: acetycholinesterase; ACh: acetylthiocholine; DTNB: 5,5′-dithiobis(2-nitrobenzoic acid); BChE: butyrylcholinesterase; BCh: butyrylthiocholine; DMPTB: 2,3-dimercapto-1-propanol tributyrate; ACE: angiotensin-converting enzyme; HHL: hippuryl-histidyl-leucine; OPA: *o*-phthaldialdehyde; pNPG: *p*-nitrophenyl-α-D-glucopyranoside; ^1^
*Electrophorus electricus* AChE (1000 units/mg); ^2^ equine serum BChE (≥10 units/mg); ^3^
*Candida rugosa* lipase (type VII, ≥700 unit/mg); ^4^ rabbit lung ACE (≥2 unit/mg); ^5^
*Saccharomyces cerevisiae* α-glucosidase (type I, ≥10 U/mg protein).

## Data Availability

Data are contained within this article.

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
