# Peer review of "Seasonal Effects on Phenolic Contents and In Vitro Health-Promoting Bioactivities of Sacred Lotus (Nelumbo nucifera)"

_plants, 2023, doi:10.3390/plants12071441_

Round 1
Reviewer 1 Report
The lotus plant Nelumbo nucifera has been consumed as food or used as traditional medicine in Asian. This plant is rich in secondary metabolites such as phenolics and alkaloids. This manuscript has conducted a large number of activity screening and found that total phenolics are the main active sites of this plant. Further identification of specific phenolic substances from the extracts will increase the novelty of the manuscript. At least, qualitative data of phenolic compounds from the extract should be provided in the supplementary file. Totally, the language of this article is well organized and the logical thinking is clear.
Author Response
Please see the attachment.
Best regards,
Nattira

Reviewer 2 Report
It is opinion of the reviewer that this paper before acceptance needs several corrections. My individual comments are listed below.
L. 2, 59 - “in vivo” should be written with italic.
L. 2 – What does it mean “economical product”.
L. 28 – It should be “The phenolic content and …”.
L. 38 – “Nelumbonaceae” not with italic.
L. 45 – The presence of nutrients in Sacred lotus should be completed.
L. 52 – It should be “high amount of …”.
Table 1 – The SD and the range of results should be presented.
Table 2, 3, 4, 5, 6, 7, 8, 9 – The results with two digitals after decimal point (mean value and SD) should be reported.
L. 125 – It should be “… radical (DPPH•) scavenging …”.
l. 150 – It should be “… results while leaf …”
Table 5 – It should be “β-Secretase …”.
L. 234 – It should be “Type 2 diabetes …”.
L. 344 – What does it mean “7-IH moiety”? Rephrasing is needed.
L. 376 – Remove underlining.
L. 413 – It should be “Folin-Ciocalteu’s phenol reagent snt gallic acid …”.
L. 413 – It should be “ … and the absobance was read at 765 nm”.
L. 416 – It should be “UV-Vis …”.
L. 425 – “-s” with lower cace letter.
L. 441 – It should be “2.3-dimercapto- …”.
L. 524 – It should be “PLoS ONE”.
Author Response

(The authors gave the same response as above.)

Reviewer 3 Report
The manuscript entitled “Seasonal Effects on Phenolic Contents and In Vitro Health-pro-2 moting Bioactivities of Sacred Lotus (Nelumbo nucifera) dealing with some secondary metabolites such as polyphenols extracted from sacred Lotus (Nelumbo nucifera) parts and their biological activities (antioxidant activities: FRAP, ORAC and DPPH) and some enzyme inhibitory activities such as Beta-secretase (BACE-1), Acetylcholinesterase (AChE), angiotensin-converting enzyme (ACE), and α-glucosidase inhibitory activities of different sacred lotus parts collected from different 243 time periods is very well described and discussed.
Even if not too original, the manuscript could be of some interest for the readers of the Journal. The technical aspects of the presented data seem to be good. In the Discussion section the authors provide a good presentation of their results. The manuscript is written in a good English. For these reasons, in my opinion the manuscript could be suitable for the publication after minor corrections:
1. In the Sample Preparation and Extraction section, please indicate the mean annual temperature, mean annual precipitation and moisture. In addition, authors must add a voucher specimen number for future plant specie identification.
2. When the work was undertaken on polyphenols extraction and determination of their biological activities, in my opinion, a sample HPLC analysis of polyphenols was required in order to see the main compounds that could be responsible of these biological activities. So, I suggest that authors analyze polyphenols and insert a chromatogram or table of their composition in different plant parts at the three mentioned times. If not possible state only the 3 major compounds from literature in the discussion section.
3. Line 419, please change the subtitle (4.3 Analysis of Antioxidant Activities) to (4.3.Antioxidant Activities) or 4.3 Determination of Antioxidant Activities
Author Response

(The authors gave the same response as above.)
